# Identification and Characterization of Three New Antimicrobial Peptides from the Marine Mollusk *Nerita versicolor* (Gmelin, 1791)

**DOI:** 10.3390/ijms24043852

**Published:** 2023-02-14

**Authors:** Armando Rodriguez, Ernesto M. Martell-Huguet, Melaine González-García, Daniel Alpízar-Pedraza, Annia Alba, Antonio A. Vazquez, Mark Grieshober, Barbara Spellerberg, Steffen Stenger, Jan Münch, Ann-Kathrin Kissmann, Frank Rosenau, Ludger A. Wessjohann, Sebastian Wiese, Ludger Ständker, Anselmo J. Otero-Gonzalez

**Affiliations:** 1Core Facility for Functional Peptidomics (CFP), Faculty of Medicine, Ulm University, 89081 Ulm, Germany; 2Core Unit of Mass Spectrometry and Proteomics, Faculty of Medicine, Ulm University, 89081 Ulm, Germany; 3Center for Protein Studies, Faculty of Biology, University of Havana, 25 and I, La Habana 10400, Cuba; 4Center for Pharmaceutical Research and Development (CIDEM), 26th Avenue, No. 1605, Nuevo Vedado, La Habana 10400, Cuba; 5Department of Parasitology, Institute of Tropical Medicine “Pedro Kouri”, Autopista Novia del Mediodía, La Habana 13600, Cuba; 6Institute of Medical Microbiology and Hygiene, University Clinic of Ulm, TBC1 Forschung, Albert-Einstein-Allee 11, 89081 Ulm, Germany; 7Institute of Molecular Virology, Ulm University, 89081 Ulm, Germany; 8Institute of Pharmaceutical Biotechnology, Ulm University, 89081 Ulm, Germany; 9Department of Bioorganic Chemistry, Leibniz Institute of Plant Biochemistry, Weinberg 3, 06120 Halle (Saale), Germany

**Keywords:** marine invertebrates, antimicrobial peptides, mass spectrometry, bioinformatical prediction, antibiotic alternatives

## Abstract

Mollusks have been widely investigated for antimicrobial peptides because their humoral defense against pathogens is mainly based on these small biomolecules. In this report, we describe the identification of three novel antimicrobial peptides from the marine mollusk *Nerita versicolor*. A pool of *N. versicolor* peptides was analyzed with nanoLC-ESI-MS-MS technology, and three potential antimicrobial peptides (Nv-p1, Nv-p2 and Nv-p3) were identified with bioinformatical predictions and selected for chemical synthesis and evaluation of their biological activity. Database searches showed that two of them show partial identity to histone H4 peptide fragments from other invertebrate species. Structural predictions revealed that they all adopt a random coil structure even when placed near a lipid bilayer patch. Nv-p1, Nv-p2 and Nv-p3 exhibited activity against *Pseudomonas aeruginosa*. The most active peptide was Nv-p3 with an inhibitory activity starting at 1.5 µg/mL in the radial diffusion assays. The peptides were ineffective against *Klebsiella pneumoniae*, *Listeria monocytogenes* and *Mycobacterium tuberculosis*. On the other hand, these peptides demonstrated effective antibiofilm action against *Candida albicans*, *Candida parapsilosis* and *Candida auris* but not against the planktonic cells. None of the peptides had significant toxicity on primary human macrophages and fetal lung fibroblasts at effective antimicrobial concentrations. Our results indicate that *N. versicolor*-derived peptides represent new AMP sequences and have the potential to be optimized and developed into antibiotic alternatives against bacterial and fungal infections.

## 1. Introduction

The emergence of microbes resistant to conventional antibiotics has become a serious health problem worldwide. According to the World Health Organization (WHO), antimicrobial resistance is one of the ten most significant threats to human health (https://www.who.int/news-room/fact-sheets/detail/antimicrobial-resistance, accessed on 13 February 2022). In this scenario, the search for new effective antimicrobial agents is mandatory.

Antimicrobial peptides (AMPs) represent a promising alternative mainly because their mode of action is substantially different from existing antibiotics, and they have a low rate of resistance development in bacteria [1]. AMPs often exhibit a broad spectrum of activity against a wide range of microorganisms. These molecules are present in many organisms, where they are an essential part of the innate immune system due to their extensive antimicrobial activity [2]. In general, they can act directly on microorganisms, but some also modulate the immune system by orchestrating a local or systemic response [3,4]. AMPs have been isolated from different sources, including plants, mammals and invertebrates. However, invertebrates are the ideal candidates since they do not have an adaptive immune system; as a result, their defenses are based primarily on a broad and effective group of AMPs that can interact directly with microbes or their toxic molecules [5]. Molluscans constitute one of the largest animal phyla with approximately 100 000 species. However, only a few of them have been explored for antimicrobial peptides since no more than 47 entries for Mollusca can be found in the AMP database (https://aps.unmc.edu/, accessed on 13 February 2022) [6]. Therefore, many new antimicrobial peptides are expected to be discovered from these organisms.

*Nerita versicolor* (Gmelin J.F.,1791; Figure 1) is a sea snail widely distributed within the shorelines of the Caribbean Sea and the Gulf of Mexico (http://www.marinespecies.org/, accessed on 13 February 2022). Although *N. versicolor* is abundant, no report has been published on the isolation and characterization of antimicrobial molecules from this species. In our study, we describe the first isolation and analysis of a pool of peptides from *N. versicolor* to identify and characterize new antimicrobial peptides.

## 2. Results

### 2.1. Peptide Sequencing Analysis with NanoLC-ESI-MS-MS

The samples obtained by the extraction of the snail *N. versicolor* were directly fractionated and analyzed with LC-MS/MS. In total, 35 peptide fragments from 10 known proteins were found with PEAKs processing of the MS raw data. From these sequences, only one was identified as a fragment from a protein annotated in the AMP database, whereas the others were from molluscan species (Appendix A). De novo sequencing yielded 125 sequences (ALC > 80%). The sequence hits were converted into the FASTA format and introduced into the AMP prediction servers CAMPR3, AMP scanner vr.2 and iAMPpred. The results were merged, and the prediction probability values from all servers were averaged for every sequence. A rank list was organized by decreasing order of prediction probability (Appendix A). The two highest-scored peptide hits from the database search list (DB + PTM + Spider) (Nv-p1: SGRGKGGKGLGKGGAKRHR and Nv-p2: SGRGKGGKGLGKGGAKRH) and the highest-scored peptide hit from the de novo list (Nv-p3: KKKPTKK) were selected for synthesis and further biological evaluation for their antimicrobial activity (Table 1).

The PEAKs search against the AMP and Mollusca databases revealed that Nv-p1 and Nv-p2 are identical to the peptide sequences of the antimicrobial histone H4 from the American cupped oyster *Crassostrea virginica*. Also, Nv-p1 and Nv-p2 are identical to the peptide sequences of histone H4 from the mussels *Mytilus galloprovincialis*, *M. californianus*, *M. edulis*, *M. trossulus*, *M. chilensis* and *M. californica* [7]. Nevertheless, these two peptides can be considered novel antimicrobial sequences, as other regions of histone H4 have been described as AMP. No similar hit was found for Nv-p3 in the Uniprot database.

### 2.2. Structural Predictions

The peptide sequence of Nv-p1 is 100% identical to the N-terminus of the histone H4-1 precursor from the African clawed frog *Xenopus laevis* (UniProt ID: AAA49736.1); therefore, its structure (PDB ID 1KX5, chain F, histone H4 from *Xenopus laevis*) was used as a template for the structural predictions. The secondary conformation predicted by PsiPred and the 3D-modeled structure were all random coil (Figure 2). The X-ray structure from the sequence identical to Nv-p1 and Nv-p2 shows 47% of the residues in the core region and 35% in the allowed region, and 18% are outliers in the Ramachandran plot. For the homology model of Nv-p1, the values are slightly better since it was modeled without the DNA with 76%, 18% and 6%, respectively. The Ramachandran plot regions are based on protein (not peptide) data, so these values might be normal for peptides.

A similar simulation with Nv-p3 was performed but did not show a preference for a helical structure. As a positive control to show that it is possible to simulate the helical content from a primary sequence, the aforementioned α-helical part of *Argopecten irradians* myosin (PDB-ID: 1B7T) was used. The 16 amino acid peptide was rebuilt from scratch and underwent the same molecular dynamics simulation protocol for 70 ns. The helical content rose to 25% at 12 ns, and later it reached 38%; therefore, simulation in a water box was sufficient to reform a part of the secondary structure. As a second positive control, the AMP magainin 2 was chosen and underwent the same procedure. Reformation of a partial helical part went much faster than with the myosin snippet; already after 1 ns, 22% helical content emerged. In both simulations, the formed helices were not stable and resolved to coil formations to be reformed later on. The stability of the helical part of myosin was tested as well; the short, cut-out helical part of the x-ray structure was observed for 100 ns.

Here, the helical content changed from 80% to 25% and back to 80%. During the simulation, five peaks with 80% helical content were counted. Therefore, reformation of a helical structure is possible, but the helical structures are dynamic under the used conditions. On the other hand, helix formation is unlikely due to high glycine content.

### 2.3. Refolding Near a Membrane

A coil-like 3D structure from Nv-p1 and Nv-p3 was placed near a phospholipid bilayer patch, and a molecular dynamic simulation was started. The secondary structures of the peptides were analyzed for up to 70 ns. No helix formation was observed.

### 2.4. Antimicrobial Activity

The antibacterial activity of these peptides was tested using radial diffusion assays (RDA) with Gram-positive and Gram-negative bacteria. None of the peptides showed activity against *K. pneumoniae* and *L. monocytogenes* at the concentrations evaluated (0.39–100 µg/mL). Figure 3 shows the antibacterial activity of all the peptides against the Gram-negative species *Pseudomonas aeruginosa*. Nv-p1, Nv-p2 and Nv-p3 showed activity at all concentrations evaluated. Nv-p3 exhibited antibacterial activity even at concentrations below 5 µg/mL. Table 2 shows the MIC determinations for Nv-p1, Nv-p2 and Nv-p3. On the other hand, all the peptides exhibited an insignificant activity against extracellular *M. tuberculosis* (Appendix A).

The antifungal activity of Nv-p1, Nv-p2 and Nv-p3 was evaluated with the broth microdilution assay using three species of *Candida*, *C. albicans*, *C. parapsilosis* and *C. auris*, under CLSI guidelines following 24 h of incubation at 37 °C (Figure 4). Also, the antibiofilm activity of these peptides was evaluated by quantification of the biofilm mass reduction. In general, the three peptides were ineffective against planktonic cells. Only Nv-p1 showed a relevant activity at the highest concentration (150 µg/mL) against planktonic *C. parapsilosis* and *C. auris*.

In contrast, the peptides exhibited a remarkable antibiofilm activity with the exception of Nv-p1 in the *C. albicans* biofilms. Nv-p2 and Nv-p3 reduced the biofilm mass to approximately 50% in the evaluated *Candida* species at all the tested concentrations, whereas Nv-p1 demonstrated a similar effect only in *C. parapsilosis*. However, Nv-p1 showed the most potent and specific antibiofilm activity of these peptides, which can be observed by the complete elimination of the *C. auris* biofilm even at the lowest concentration of 5 µg/mL. The MIC and IC50 of the antifungal and antibiofilm activities against every strain were calculated and are shown in Table 3.

### 2.5. Toxicity Assays

The in vitro toxicity of the Nv-p (1–3) peptides was evaluated using the human fetal lung fibroblast cell line WI-38 and primary human macrophages (Figure 5). Nv-p2 and Nv-p3 slightly reduced the viability of the WI-38 cells at the highest concentration of 100 µg/mL, but not at the other concentrations.

The results show that the cell viability was not reduced below 80% by any of the peptides in the primary human macrophages (Table 4).

## 3. Discussion

The alarming increase in bacterial strains resistant to conventional antibiotics has become a serious health problem worldwide. To address this problem, the search for and development of new antimicrobial molecules with novel mechanisms of action are mandatory. In this context, AMPs represent exciting candidates due to their broad antimicrobial spectra and the difficulty for pathogens to develop resistance to them [8].

AMPs are present in all living organisms as part of the innate immune system, but in invertebrates, they represent the main component of humoral immunity; thus, they are excellent candidates to be explored in the search for AMPs [9]. Therefore, we focused on the analysis of the unexplored marine mollusk *N. versicolor* to identify new antimicrobial peptides with LC-MS/MS and in silico prediction of antimicrobial activity. Among the AMP candidates, we selected the three most promising ones (Nv-p1 and Nv-p2 from the database search, and Nv-p3 from de novo sequencing) for synthesis and further biological evaluation. To our knowledge, these are the first antimicrobial peptides identified in the molluscan species *N. versicolor*.

The Nv-p1 and the Nv-p2 sequences detected with LC-MS/MS in the Nerita peptidome show N-terminal acetylation, which we can see in the PEAKS software. Since the prediction servers only allow import sequences without PTMs, we first started synthesis and testing of the unmodified peptides without PTMs. Although many AMPs have been identified in molluscans, histone-like peptides only represent a small group [10]. Histones are among the most conserved eukaryotic proteins, being very similar even in organisms from different Phyla. This fact suggests that most residues are essential for their function, which comprises the association with DNA in the nucleus to form chromatin together with the non-histone chromosomal proteins [11]. As previously suggested, histones played an important role in ancient host defense systems against pathogenic microorganisms prior to their integration as elements of chromatin structure in eukaryotes [12]. In the last few decades, naturally occurring antimicrobial histone-like peptides have been reported [13,14]. Some of these peptides share high sequence identity to N-terminal fragments of human histone H2A, such as parasin I (19 aa residues), buforin I (39 aa residues) and hipposin I (51 aa residues).

On the other hand, natural histone H3 fragments have not been reported, and the studies of antimicrobial histone H4-like fragments have been focused on the 86–100 region [15]. In the present study, we found naturally occurring histone H4 fragments (Nv-p1 and Nv-p2) involving a different region, the N-terminus (histone H4, 2–19 and 2–20), which so far was not found in nature and has not been described as an antimicrobial region in histone H4. The antimicrobial activity of full-length histone H4 sequences, e.g., from human sebocytes, was described against *Staphylococcus aureus* and *Propionibacterium acnes* [16]. Another full-length, oyster-derived histone H4 was tested against *Vibrio anguillarum* and *Escherichia coli* [7]. Both tested histone H4 peptides showed antimicrobial activity against Gram+ and Gram- bacteria in similar micromolar concentrations if compared to our N-terminal and smaller histone H4 sequences.

Contrasting to H2A (also H2A-like peptides) and H2B, which penetrate the cell membrane to bind bacterial DNA, histone H4 exerts its antimicrobial activity by destroying the bacterial cell membrane [17]. However, histogranin, which is identical to the C-terminus of histone H4 (86–100), is believed to act at an intracellular level that is ATP-dependent [15]. A study of intracellular-targeting antimicrobial peptides using *Escherichia coli* proteome microarrays demonstrated that few individual AMPs can interact with multiple intracellular targets, limiting the development of bacterial resistance given the complexity of their antimicrobial mechanisms [18]. In this sense, given that the histone family is known to yield intracellular targeting AMPs, it would be interesting to study the mechanism of action of Nv-p1 (2–19) and Nv-p2 (2–20). Similar to other histone-like AMPs, Nv-p1 and Nv-p2 include the histone N-terminus, which contains the DNA-binding region; therefore, bacterial DNA/RNA could be one of the targets if its antibacterial activity was intracellular.

Regarding Nv-p3, no identical sequences are found in the Uniprot database. However, Nv-p3 shares similarity with some peptide toxins, having in common affinities for cell membranes due to the combination of basic and hydrophobic residues. Ion channel toxins bind to the cell membrane to modulate the ion channel’s activity, whereas phospholipase A2 causes cell membrane damage. The antimicrobial activities of animal venoms and toxins are documented [19,20,21].

To predict the conformation of the Nv peptides, we employed homology modeling after having found an adequate template with more than 25% sequence identity [22]. Nv-p1 and Nv-p2 showed 100% sequence identity with the N-terminus of histone H4 from Tetrahymena Gcn5. This N-terminal part of H4 associates with DNA and shows a specific conformation that is not helical, as shown by the predicted secondary structure of Nv-p1. Nv-p3 demonstrated the same behavior. A unique extended coil structure is one of the three structural subclasses of AMPs. Most AMPs in this category contain a high content of arginine, proline, tryptophan and histidine residues. The best-studied AMPs from this group are from the cathelicidin family [8,23].

The antibacterial activity of the Nv-p (1–3) peptides was evaluated using the agar diffusion assay against Gram-negative and Gram-positive bacteria. These strains are not multidrug-resistant microorganisms, but it is important to note that all of them are opportunistic pathogens that can cause severe infections. None of the peptides showed activity against *K. pneumoniae* and *L. monocytogenes* at the concentrations evaluated. The Nv-p (1–3) peptides exhibited action against *P. aeruginosa* at all the concentrations evaluated. However, Nv-p3 was the most potent peptide with antibacterial action at concentrations below 5 µg/mL.

*Mycobacterium tuberculosis* is a causal agent of tuberculosis, a re-emerging disease. In recent years, it has received significant attention due to the appearance of multiresistant strains to conventional antituberculosis drugs. In this sense, we evaluated the Nv-p (1–3) peptides against extracellular *M. tuberculosis*, and we noted that the peptides exhibited insignificant antimycobacterial activity (Appendix A).

The antifungal activity of the Nv-p (1–3) peptides was evaluated through the broth microdilution assay. None of these peptides showed relevant action against planktonic *Candida spp* except Nv-p1 at the highest concentration of 150 µg/mL. On the other hand, these peptides demonstrated an effective antibiofilm activity in all the concentrations evaluated, but in this case, Nv-p1 does not show biofilm mass reduction in *C. albicans*. The effect of Nv-p2 and Nv-p3 was the same against the three *Candida* species in all the evaluated concentrations suggesting the broad spectrum of antibiofilm activity of these peptides. Nv-p1 reduced the biofilm mass to approximately 50% in *C. parapsilosis* and completely in *C. auris*. In this sense, Nv-p1 exhibited the most potent and specific antibiofilm activity of these peptides. Regarding histone-like peptides, Nv-p1 is the first known natural histone H4 fragment found to exhibit such a powerful antibiofilm activity.

Toxicity is one of the main disadvantages of the therapeutic application of AMPs [24]. Therefore, the toxicity of the Nv-p (1–3) peptides was evaluated using human primary macrophages and the human fetal lung fibroblast cell line WI-38. These cells have been previously used as in vitro cellular models for testing the toxicity of antimicrobial peptides [24,25]. Also, they have been proven to be sensitive enough to assess the availability of these molecules as potential anti-infectious drugs in humans [25]. None of the Nv-p (1–3) peptides exhibited significant toxic effects against human cells. Only Nv-p2 and Nv-p3 showed a decrease in cell viability below 80% at 100 µg/mL using the WI-38 cells. This concentration is higher than needed for antimicrobial activity; for that reason, we expect that the effective concentrations of these peptides are not toxic.

According to WHO, antimicrobial resistance is one of the top 10 global public health threats facing humanity; therefore, the search for novel antimicrobial molecules is of great interest. Given the large variety of molluscan species, they represent a rich source of novel peptides to be used as antimicrobials.

Our results indicate that the Nv-p (1–3) peptides have the potential to be optimized and developed into appropriate therapeutic candidates as antibiotic alternatives or complements against bacterial and fungal infectious agents.

## 4. Materials and Methods

### 4.1. Invertebrate Collection and Sample Preparation

We collected eight individuals of marine mollusk *N. versicolor* at Jibacoa Beach, East Havana, Cuba. They were homogenized in saline phosphate buffer (PBS) in a blender. The homogenate obtained was centrifuged at 10,000 rpm for 15 min at 4 °C, and the supernatant was ultrafiltrated with a molecular weight cut-off of 10 kDa at 5371 rpm for 10 min at 4 °C. The low-molecular-weight fraction was lyophilized and stored at −20 °C before fractionation and analysis with LC-MS/MS.

### 4.2. Microorganism Strains and Growth Conditions

Four bacterial species and one yeast species were used to evaluate the antimicrobial activity of the putative AMPs. *Pseudomonas aeruginosa* ATCC 29213, *Listeria monocytogenes* ATCC BAA-679/EGD-e and *Klebsiella pneumoniae* ATCC 70,063 were cultured at 37 °C in a 5% CO_2_ atmosphere overnight as preculture inoculum in Todd–Hewitt broth. Yeast *Candida albicans* (ATCC 90028) was grown in Sabouraud dextrose medium at 37 °C for 48 h as preculture inoculum. *Mycobacterium tuberculosis* ATCC 27,294 was grown in 7H9 medium containing 7H9 BBL Middlebrook broth (BD), glycerol, OADC (Oelic Albumin Dextrose Catalase; BD), Tween 80 and ddH_2_O. The pH was adjusted between 7.2–7.4, and sterile filtration was performed with a 0.2 µm filter membrane (Thermo ScientificTM NalgeneTM Rapid-FlowTM; Thermo Fisher Scientific GmbH, Dreieich, Germany).

### 4.3. Human Cells and Culture Conditions

Human monocyte-derived macrophages (hMDMs): PBMCs were isolated from human buffy coat via high-density gradient centrifugation (Ficoll-Paque Plus; GE Healthcare Germany, Dornstadt, Germany) (Ethical vote from the University of Ulm no. 93/21). The monocytes were then purified from the PBMCs through adherence. The cells were stimulated with GM-CSF (10 ng/mL; Miltenyi Biotech, Bergisch Gladbach, Germany) in macrophage serum-free medium (Gibco) for six days at 37 °C and a 5% CO_2_ atmosphere. The cell line WI-38 (ATTC number CCL-75) of human fetal lung fibroblasts was cultured in Dulbecco’s modified Eagle’s medium containing 10% fetal bovine serum, 50 U/mL penicillin and 50 mg/mL streptomycin (Gibco, Gaithersburg, MD, USA) and incubated under a 5% CO_2_ atmosphere at 37 °C.

### 4.4. Peptide Sequencing Analysis with NanoLC-ESI-MS-MS

Land snail samples were reduced with 5 Mm DTT for 20 min at RT and subsequently alkylated with iodoacetamide for 20 min at 37°. The samples (15 µL) were analyzed using an LTQ Orbitrap Velos Pro system (Thermo Fisher Scientific) online coupled to a U3000 RSLCnano (Thermo Fisher Scientific) Uplc as described previously [26].

De novo sequencing and database searches were performed with PEAKs X software (http://www.bioinfor.com/peaks-software/, accessed on 13 February 2022) using its current workflow comprising de novo sequencing as well as database searches with user-defined modifications (PEAKs DB), post-translational modifications (PEAKs PTM) from the Unimod database “http://www.unimod.org/modifications_list.php accesed on 13.02.2023” and mutations (Spider). For peptide identification, the MS/MS spectra were correlated with the UniProt reviewed Mollusca proteins (http://www.uniprot.org, accessed on 13 February 2022) and the APD3 antimicrobial peptide database (http://aps.unmc.edu/AP/, accessed on 13 February 2022) [6]. For the database search with PEAKs DB, carbamidomethylated cysteine was considered as a fixed modification along with oxidation (M) as a variable modification. False discovery rates were set on both peptide and protein levels to 1%. On the other hand, only those de novo peptides with an average local confidence higher (ALC) than 80% were selected.

### 4.5. Antimicrobial Activity Prediction

The peptide list exported from the PEAKs analysis was converted into a FASTA file and introduced in the following antimicrobial peptide prediction servers: CAMPR3, (http://www.camp3.bicnirrh.res.in/predict, accessed on 13 February 2022) [27], AMP scanner vr.2, (https://www.dveltri.com/ascan/v2/ascan.html, accessed on 13 February 2022) [28] and iAMPpred (http://cabgrid.res.in: 8080/amppred/server.php accessed on 13 February 2022) [29].

### 4.6. Peptide Synthesis

Peptides were synthesized (Fmoc technique) and analyzed with LC-MS (electrospray) and analytical HPLC (Appendix A) as previously described [30]. The stock solutions of the peptides were formulated by weighing the lyophilized peptides. The water content of the lyophilized peptides was corrected with amino acid analysis using an ACQUITY UPLC H-Class-Amino-Acid-Analyzer (Waters Corporation, Milford, MA, USA).

### 4.7. Structural Prediction

Calculation of the 3D structures from the peptide sequences was performed with homology modeling followed by ab initio folding in water using molecular dynamics simulations. Regarding this, using each peptide sequence as a query, a simple protein BLAST search was performed in all nonredundant databases using a BLOSUM62 matrix. The homology modeling was carried out using Modeller v10.2. The obtained models were refined by 100 ns of molecular dynamics simulations using the forcefield CHARMM36 implemented in the software NAMD. All models were solvated with a cubic water box with the water model TIP3P, and ions were added to neutralize the systems. The water box was added at 15 Å from the edges of the protein. The model quality was assessed using the Ramachandran plot and Prosa.

For estimating their behavior near membranes, the 3D peptide structures were modeled in water at 5 Å from a lipid bilayer patch. This membrane patch consisted of three membrane lipids, phosphatidylethanolamine, phosphatidylcholine and phosphatidylserine, all with 1-palmitoyl and 2-oleyl side chains. The membrane was built using the CHARMM-GUI server (https://www.charmm-gui.org/, accessed on 13 February 2022), and the peptides were manually located at 5 Å from the membrane. The molecular dynamics simulations were performed under the same condition as before but for 70 ns.

### 4.8. Determination of Antimicrobial Activity

Susceptibility of *Pseudomonas aeruginosa* strain ATCC27853 to Nv-p1, Nv-p2 and Nv-p3 peptides was tested with the radial diffusion assay (RDA). Bacteria were cultured in liquid broth at 37 °C in a 5% CO_2_ atmosphere overnight, pelleted with centrifugation and washed in 10 mM sodium phosphate buffer. Following resuspension in 10 mM sodium phosphate buffer, the optical density was determined at 600 nm. A total of 2 × 10^7^ bacteria were seeded into a petri dish in 1% agarose and 10 mM sodium phosphate buffer. After cooling at 4 °C for 30 min, 3–5 mm holes were cut into the 1% agarose. The peptides, adjusted to the indicated concentration in 10 µL of buffer, were filled into the agar holes. Following an incubation at 37 °C in ambient air for 3 h, the plates were overlaid with 1% agarose and 3% tryptic soy solved in 10 mM phosphate buffer. The inhibition zones in cm were determined following 16–18 h incubation time at 37 °C and a 5% CO_2_ atmosphere. Determinations were repeated five times and depicted as mean values and standard deviations. LL37 at a concentration of 100 µg/mL served as the positive control. In the inhibition zones of Nv-p1, Nv-p2 and Nv-p3, single colonies were sometimes present.

The bacterial MIC determinations were performed in Mueller-Hinton broth in accordance with CLSI guidelines following overnight incubation at 37 °C. The OD 600 nm of an overnight culture of *Pseudomonas aeruginosa* was determined and adjusted to 0.1 in Mueller–Hinton broth and, afterwards, diluted 1:4. A total volume of 5 μL of this bacterial solution was mixed with 95 μL of Mueller–Hinton broth containing Nv-p1, 2 or 3, resulting in final peptide concentrations from 100 to 0.195 μg/mL. After a 24 h incubation at 37 °C and a 5% CO_2_ atmosphere, OD 600 nm was determined using a Tecan microplate reader M infinite 200. All tests were performed in triplicate.

### 4.9. ^3^H-Uracil Proliferation Assay

Extracellular *Mycobacterium tuberculosis* bacilli (2 × 10^6^) were distributed into 96-well flat bottom plates (Nunc) in triplicate and incubated for 3 days at 37 °C/5% CO_2_. Next, ^3^H-Uracil (0.3 µCi/mL, Biotrend Chemikalien GmbH, Köln, Germany) was added overnight at 37 °C/5% CO_2_. The bacteria were then fixed and killed with 4% paraformaldehyde (PFA) at room temperature (RT) for 20 min and then harvested (Cell harvester; Inotech Bioscience LLT, Derwod, MD, USA) onto a filtermat (Perkin Elmer, Waltham, MA, USA). Afterward, wax plates (Meltilex A; Perkin Elmer) containing scintillation liquid were molten onto the mats. The samples were measured with a beta counter (Hidex sense micro beta counter), and the mean values of the triplicates were calculated [30].

### 4.10. Antifungal Bioassays

The antifungal activity was determined according to the “Clinical and Laboratory Standards Institute” guidelines M27-A3 broth microdilution assay with modifications (turbidimetric detection) [29,31]. Based on the cell density measurements, the minimal inhibitory concentration (MIC) was derived from a Lambert–Pearson plot [32]. Flat-bottomed sterile 96-well plates (SARSTEDT, AG & Co KG, Nümbrecht, Germany), RPMI 1640 without sodium bicarbonate and MOPS buffer (Sigma-Aldrich-Merck, Darmstadt) were used for the test, and readings were performed at λ = 600 nm.

### 4.11. Biofilm Formation and Quantification/AntiBiofilm Treatment

Biofilms were basically formed and analyzed as described previously [31,32,33]. In brief, 2.5 × 10^3^ yeast cells were seeded in 200 µL RPMI-1640 medium supplemented with L-glutamine in flat-bottomed, 96-well polystyrene microtiter plates (Sarstedt AG & Co. KG, Nümbrecht, Germany) and incubated at 37 °C without agitation for 24 h. The effect of the peptides on the biofilm formation was tested at different concentrations. The biofilm was quantified with a crystal violet assay, which was originally developed for bacteria by George O’ Toole [33,34] and is also widely used for *Candida* biofilms [32,35,36]. Planktonic cells were removed with the supernatant, and the mature biofilms were washed twice with 200 µL water. Subsequently, the biofilms were stained with 200 µL of a 0.1% (*w*/*v*) crystal violet solution for 15 min. The supernatant was removed, and the biofilms were washed twice with 200 µL water to eliminate excess crystal violet. The stained biofilms were air dried for 24 h at 25 °C and finally destained using 200 µL of 30% acetic acid (15 min, 25 °C). The supernatant was transferred to a fresh 96-well plate, and the absorbance at 560 nm was measured using a Tecan infinite M200 microplate reader to quantify the biofilm biomass. The curve was fitted with a nonlinear regression with the dose-response nonlinear Hill equation. The semi-inhibitory concentration of biofilm formation (IC50b) represents the point at which the biofilm mass is reduced to 50% compared to the biofilm mass of the untreated control.

### 4.12. Toxicity Assays

A total of 1 × 10^5^ macrophages or WI-38 cells per well were distributed into a 96-well plate in corresponding cell culture medium. The controls were the media control, heat-inactivated cells and a diluent control (sterile water; Fresenius Kabi, Bad Homburg, Germany). The cells were then stimulated with the peptides at different concentrations overnight. Twenty microliters of PrestoBlue Cell Viability Reagent (Life Technologies, Darmstadt, Germany) was added per well and incubated for 20 min at 37 °C and a 5% CO_2_ atmosphere. Fluorescence was measured at 560 nm (excitation) and 600 nm (emission) with a TECAN infinite M200 microplate reader. The OD of the media control was then subtracted from the other results. The unstimulated cells were set to 100% viability [30].

## Figures and Tables

**Figure 1 ijms-24-03852-f001:**
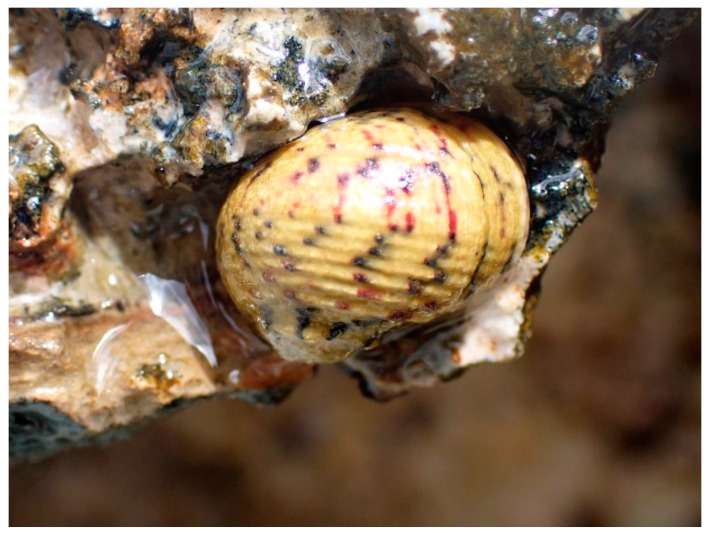
*Nerita versicolor* (Gmelin J.F.,1791). Original, unmodified photograph from Pieter Bruijsten (https://observation.org/photos/45117118/, accessed on 13 February 2022), License: CC-BY-NC-ND (Creative Commons (https://creativecommons.org/licenses/by-nc-nd/4.0/deed.de, accessed on 13 February 2022).

**Figure 2 ijms-24-03852-f002:**
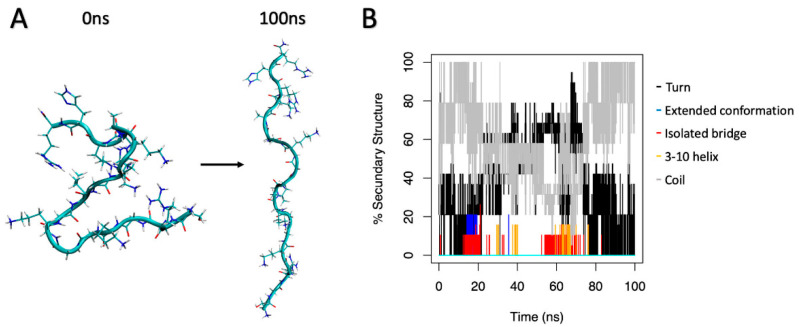
(**A**) 3D structure of Nv-p1 obtained from homology modeling (0 ns) and after 100 ns of molecular dynamics simulations. (**B**) Percentage of secondary structures of Nv-p1 through the whole molecular dynamics simulation.

**Figure 3 ijms-24-03852-f003:**
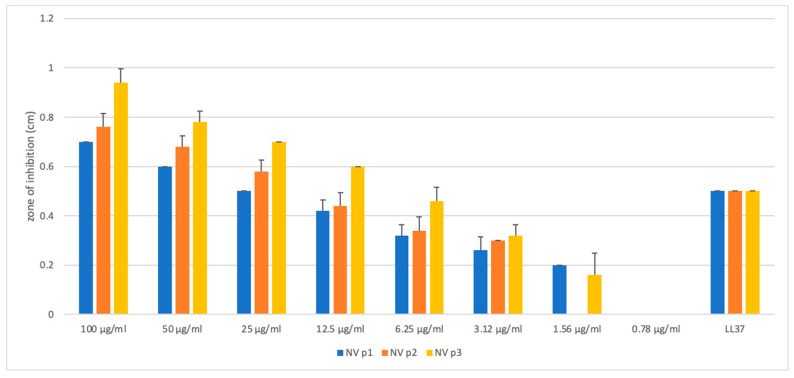
Antibacterial activity of Nv-p1, Nv-p2 and Nv-p3 peptides against *P. aeruginosa* at different concentrations (0.78–100 µg/mL) using the radial diffusion assay. Experiments were performed in quintuplicates, and the results are shown as the mean values of three independent experiments with their corresponding standard deviations.

**Figure 4 ijms-24-03852-f004:**
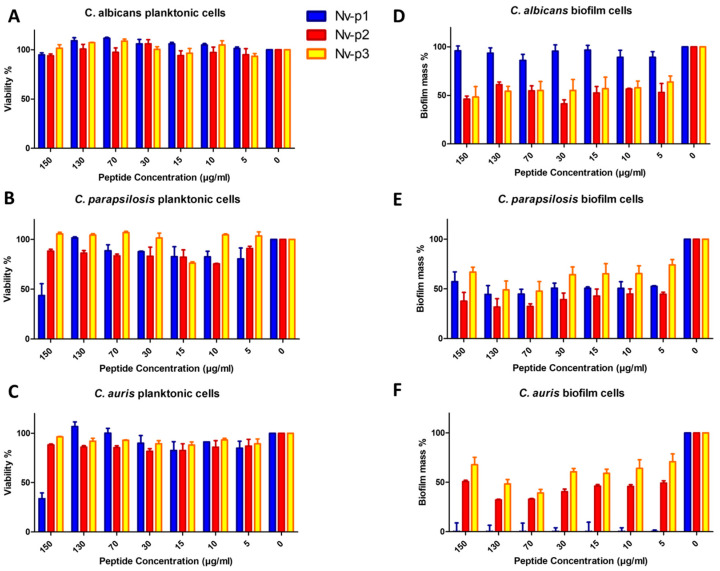
Antifungal activity of the Nv-p (1–3) peptides against planktonic cells of *C. albicans* (**A**), *C. parapsilosis* (**B**) and *C. auris* (**C**) evaluated with the broth microdilution assay at different concentrations (5–150 µg/mL). (**D**–**F**) Antibiofilm activity of these peptides evaluated by quantification of the biofilm mass reduction in the same conditions. Both tests were performed in triplicate, and the results are shown as the mean values of the three independent experiments with the corresponding standard deviations.

**Figure 5 ijms-24-03852-f005:**
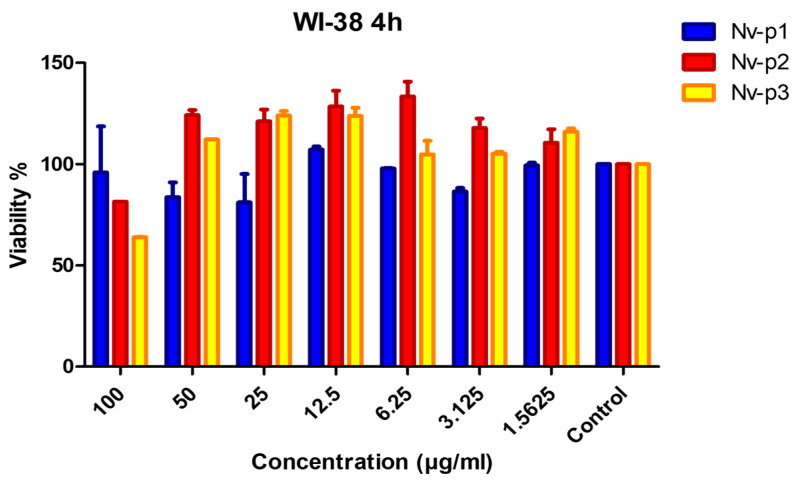
In vitro cytotoxicity of the Nv-p (1–3) peptides in the WI-38 cells at different concentrations (0.39–100 µg/mL). The viability percentage was determined with the PrestoBlue assay at 24 h of incubation with the peptides at 37 °C. Untreated cells were used as the negative control or 100% viability control. This experiment was performed in triplicate, and the results are shown as the mean values of the three independent experiments.

**Table 1 ijms-24-03852-t001:** Primary structure, pI, charge and molecular weight of Nv-p1, Nv-p2 and Nv-p3. Calculations were performed with Pro pi, https://www.protpi.ch/ (accessed on 13 February 2022).

Peptide	Primary Structure	pI	Charge at pH 7.4	Molecular Weight
Nv-p1	SGRGKGGKGLGKGGAKRHR	12.57	+6.25	1864.127 Da
Nv-p2	SGRGKGGKGLGKGGAKRH	12.51	+5.44	1707.941 Da
Nv-p3	KKKPTKK	10.57	+4.141	857.0973 Da

**Table 2 ijms-24-03852-t002:** MIC determinations of Nv-p1, Nv-p2 and Nv-p3 peptides against *P. aeruginosa*.

Peptide Bacterial species	Nv-p1	Nv-p2	Nv-p3
*Pseudomonas aeruginosa*	>100 µg/mL	>100 µg/mL	>100 µg/mL

**Table 3 ijms-24-03852-t003:** Antifungal and antibiofilm activity of the Nv-p (1–3) peptides against *Candida albicans*, *Candida parapsilosis* and *Candida auris* and their respective MIC and IC50. All tests were performed in triplicate, and the results are shown as the mean values of the three independent experiments.

Peptides	Antifungal Activity MIC (µg/mL)			Antibiofilm Activity IC50 (µg/mL)		
	*C.albicans*	*C.parapsilosis*	*C.auris*	*C.albicans*	*C.parapsilosis*	*C.auris*
Nv-p1	˃100	˃100	˃100	˃100	31.9	1.8
Nv-p2	˃100	˃100	˃100	40.0	12.4	15.4
Nv-p3	˃100	˃100	˃100	57.5	84.4	57.5

**Table 4 ijms-24-03852-t004:** In vitro cytotoxicity of the Nv-p (1–3) peptides at 100 µg/mL on primary human macrophages. The experiments were performed in duplicate, and the results are shown as the mean values of the two independent donors.

Peptide	Viability (%)
Unstimulated	100
Nv-p1	100
Nv-p2	100
Nv-p3	86

## Data Availability

Original data are provided by the corresponding authors on request.

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
