# Peer review of "Identification and Characterization of Three New Antimicrobial Peptides from the Marine Mollusk Nerita versicolor (Gmelin, 1791)"

_ijms, 2023, doi:10.3390/ijms24043852_

Round 1

Reviewer 1 Report

This paper “Identification and characterization of three novel antimicrobial peptides from the marine mollusk Nerita versicolor (Gmelin, 1791)” by Armando Rodriguez et al have identified and characterize 3 novel antimicrobial peptides with significant activities. The overall content of this MS is sufficient to publish in IJMS.

The only concern I have is how author has made the stock concentration of peptide as the designed peptides has no aromatic amino acids like tryptophan or tyrosine. The author should mention and describe it clearly in the materials and methods section.

Reviewer 2 Report

The article "Identification and characterization of three novel antimicrobial 

peptides from the marine mollusk Nerita versicolor (Gmelin 1791)" is a well-written manuscript. 

Discussion can be improved. 

There are multiple opportunistic bacteria out there. Why did the authors select only Klebsiella pneumonia and Listeria monocytogenes?

You have performed an assay with Mycobacterium tuberculosis, but the data has not been shown. It would be good if you could include the data in the article.

As AMPs are designed or gained importance because of their ability to act against the MDR organisms, it would have been more beneficial if authors considered checking against the multiple MDR strains. 

Your peptides are structurally related to H4 peptides and I only see the discussion on it; It's better to discuss another H4 kind of peptides reported previously and their antimicrobial characteristics in the discussion.

Reviewer 3 Report

In this manuscript, Rodriguez and co-authors collected samples identified three peptides from the sea snail Nerita versicolor and tested the antimicrobial activity with the synthetic version. Overall, I think this is a well-written manuscript. However, there are still a few places that may require attention.

I understand that the authors tested the antimicrobial activities and think these peptides may be AMPs. But have the authors considered these peptides have other functions or simply just degradation from the animal proteome?

The predicted peptides have PTMs on them. Could you specify if the synthetic peptides also have those PTMs, or just the unmodified version? Please show more raw data on the peptide synthesis. For example, the Nv-p3 has many Lysine, and I’d like to see the purity/quality of the peptide. If the unmodified peptides were made, then why not use the PTM version?

Please provide more rationales for investigating the peptide 3D structure. For example, how does this information help to understand the antimicrobial activity?

Are the data in Table2 correct? Doesn’t seem to correlate with the data in Figure3?

Minor issues:

Title: I would suggest changing the word “novel” to “new”

Line 67: the link provided is not openable

Line 150: Figure 3, the x-axis label the numbers “,” -> “.”

Line 412: 10^3
